# Synergistic Interferon-Alpha-Based Combinations for Treatment of SARS-CoV-2 and Other Viral Infections

**DOI:** 10.3390/v13122489

**Published:** 2021-12-11

**Authors:** Aleksandr Ianevski, Rouan Yao, Eva Zusinaite, Laura Sandra Lello, Sainan Wang, Eunji Jo, Jaewon Yang, Erlend Ravlo, Wei Wang, Hilde Lysvand, Kirsti Løseth, Valentyn Oksenych, Tanel Tenson, Marc P. Windisch, Minna M. Poranen, Anni I. Nieminen, Svein Arne Nordbø, Mona Høysæter Fenstad, Gunnveig Grødeland, Pål Aukrust, Marius Trøseid, Anu Kantele, Eglė Lastauskienė, Astra Vitkauskienė, Nicolas Legrand, Andres Merits, Magnar Bjørås, Denis E. Kainov

**Affiliations:** 1Department of Clinical and Molecular Medicine (IKOM), Norwegian University of Science and Technology, 7028 Trondheim, Norway; aleksandr.ianevski@ntnu.no (A.I.); rouan.yao@ntnu.no (R.Y.); erlend.ravlo@ntnu.no (E.R.); wei.wang@ntnu.no (W.W.); hilde.lysvand@ntnu.no (H.L.); kirsti.loseth@ntnu.no (K.L.); valentyn.oksenych@ntnu.no (V.O.); svein.a.nordbo@ntnu.no (S.A.N.); mona.hoyseter.fenstad@stolav.no (M.H.F.); magnar.bjoras@ntnu.no (M.B.); 2Institute of Technology, University of Tartu, 50411 Tartu, Estonia; eva.zusinaite@ut.ee (E.Z.); laura.lello@ut.ee (L.S.L.); sainan.wang@ut.ee (S.W.); tanel.tenson@ut.ee (T.T.); andres.merits@ut.ee (A.M.); 3Applied Molecular Virology Laboratory, Institut Pasteur Korea, Seongnam-si 463-400, Gyeonggi-do, Korea; eunji.jo@ip-korea.org (E.J.); jaewon.yang@ip-korea.org (J.Y.); marc.windisch@ip-korea.org (M.P.W.); 4Molecular and Integrative Biosciences Research Programme, Faculty of Biological and Environmental Sciences, University of Helsinki, 00014 Helsinki, Finland; minna.poranen@helsinki.fi; 5Institute for Molecular Medicine Finland, University of Helsinki, 00014 Helsinki, Finland; anni.nieminen@helsinki.fi; 6Department of Medical Microbiology, St. Olavs Hospital, 7006 Trondheim, Norway; 7Department of Immunology and Transfusion Medicine, St. Olavs Hospital, 7006 Trondheim, Norway; 8Research Institute of Internal Medicine, Oslo University Hospital Rikshospitalet, 0372 Oslo, Norway; gunnveig.grodeland@medisin.uio.no (G.G.); paukrust@ous-hf.no (P.A.); marius.troseid@medisin.uio.no (M.T.); 9Institute of Clinical Medicine (KlinMed), University of Oslo, 0318 Oslo, Norway; 10Section of Clinical Immunology and Infectious Diseases, Oslo University Hospital Rikshospitalet, 0372 Oslo, Norway; 11Inflammation Center, Infectious Diseases, Helsinki University Hospital, 00029 Helsinki, Finland; anu.kantele@helsinki.fi; 12Life Sciences Center, Vilnius University, 10257 Vilnius, Lithuania; egle.lastauskiene@gf.vu.lt; 13Department of Laboratory Medicine, Lithuanian University of Health Science, 44307 Kaunas, Lithuania; astra.vitkauskiene@kaunoklinikos.lt; 14Oncodesign, 25 Avenue du Québec, 91140 Villebon Sur Yvette, France; nlegrand@oncodesign.com

**Keywords:** interferon-alpha, antiviral drug combination, SARS-CoV-2, hepatitis C virus, hepatitis E virus, influenza A virus, human immunodeficiency virus

## Abstract

Background: There is an urgent need for new antivirals with powerful therapeutic potential and tolerable side effects. Methods: Here, we tested the antiviral properties of interferons (IFNs), alone and with other drugs in vitro. Results: While IFNs alone were insufficient to completely abolish replication of severe acute respiratory syndrome coronavirus 2 (SARS-CoV-2), IFNα, in combination with remdesivir, EIDD-2801, camostat, cycloheximide, or convalescent serum, proved to be more effective. Transcriptome and metabolomic analyses revealed that the IFNα–remdesivir combination suppressed SARS-CoV-2-mediated changes in Calu-3 cells and lung organoids, although it altered the homeostasis of uninfected cells and organoids. We also demonstrated that IFNα combinations with sofosbuvir, telaprevir, NITD008, ribavirin, pimodivir, or lamivudine were effective against HCV, HEV, FLuAV, or HIV at lower concentrations, compared to monotherapies. Conclusions: Altogether, our results indicated that IFNα can be combined with drugs that affect viral RNA transcription, protein synthesis, and processing to make synergistic combinations that can be attractive targets for further pre-clinical and clinical development against emerging and re-emerging viral infections.

## 1. Introduction

Viral diseases continue to pose a serious threat to public health, due to a paucity of effective, rapidly deployable, and widely available control measures [1,2,3]. When viruses infect the cells, viral pathogen-associated molecular patterns (PAMPs) are recognized, and signals are transduced to activate intrinsic and extrinsic immune responses [4]. Pattern recognition receptors (PRRs) sense incoming viruses, and activate transcription of certain IFN genes via NF-kB and other pathways. IFNs launch expression of IFN-stimulated genes (ISGs) in infected cells, as well as in nearby non-infected cells, protecting them from potential viral invasion. This activation of the innate immune response, combined with contributions from the adaptive immune response in the host, are often sufficient for elimination of most viruses [5].

IFNs are a large class of host proteins that are classified according to the cellular receptor to which they bind [6,7,8] (Appendix A). Type I IFNs consist of IFN-alpha (IFNα), IFN-beta, IFN-epsilon, IFN-kappa, and IFN-omega, and bind to the IFN-alpha/beta receptor IFNAR1/2. Type II IFNs consist of IFN-gamma, and interact with the IFN-gamma receptor IFNGR1/2. Type III IFNs, consisting of IFN-lambda-1-4, activate downstream signaling by forming a receptor complex, together with interleukin 10 receptor 2 and the IFN-lambda receptor [9].

Type I IFNs induce transcription of different sets of ISGs, which participate in intrinsic antiviral and extrinsic immune responses. For example, ISGs like IFIT1 and OASL activate ribonuclease L (RNaseL), which leads to the degradation of viral RNA [10]. Moreover, ISGs such as interleukins, and C-X-C and C-C motif chemokines (CXCLs and CCLs) recruit immune cells to the site of infection. Notably, mutations in IFN-signaling pathway genes have resulted in increased susceptibility to viral infections and reduced patient survival [11,12,13,14]. However, the exact role of each IFN pathway and their crosstalk remain unclear.

The use of recombinant human IFNs (rhIFNs) has been approved for treatment of hepatitis C virus (HCV) and hepatitis B virus (HBV) infections [15,16]. Additionally, IFNs have been shown to be effective against a variety of other viruses, including SARS-CoV-2 in clinical trials and in laboratory settings (Appendix A) [17,18,19,20,21]. Unfortunately, rhIFNs possess limited efficacy when administered as antiviral treatments [22,23], and can cause adverse effects when used at established doses [24].

The toxicity of rhIFNs can be reduced by combining them with other antiviral drugs that act synergistically, thus allowing for the use of smaller doses of each component (Appendix A). Moreover, synergistic combinations can often have higher efficacy against viral infections than individual components administered as monotherapies, even at lower doses [25]. Indeed, the combination of IFNα and ribavirin has been approved for treatment of chronic HCV infection for more than a decade [15]. Similarly, several rhIFN-based drug combinations have been tested against COVID-19. Of note, IFNb1b–lopinavir–ritonavir–ribavirin, IFNα2b–IFNg, and IFNα–umifenovir were all shown to be effective for treatment of patients with COVID-19 [26,27,28,29]. However, despite these promising data, the mode in which IFNs can be optimally combined with other drugs to maximize antiviral efficacy and minimize side effects remains unclear.

Here, we report several synergistic IFNα-based combination therapies that have better efficacy and lower toxicity than single drugs against SARS-CoV-2, HCV, HEV, FluAV, and HIV-1 infections in vitro. These combinations have powerful treatment potential, which can be leveraged for use in response to wild type viruses and their immune-evading or drug-resistant variants.

## 2. Materials and Methods

### 2.1. Drugs, Viruses, Cells, and Lung Organoids

All experiments with viruses were performed in BSL2 or BSL3 laboratories, in compliance with the guidelines of the national authorities, and under appropriate ethical and safety approvals. Standard operational procedures were approved by the institutional safety committee.

Appendix A lists IFNs and other antiviral agents, their suppliers, and catalogue numbers. Lyophilized IFNs were dissolved in sterile deionized water to obtain 200 μg/mL concentrations. Compounds were dissolved in dimethyl sulfoxide (DMSO; Sigma-Aldrich, Hamburg, Germany) or Milli-Q water to obtain 10 mM stock solutions. The convalescent serum (G614) from a recovered COVID-19 patient has been described in a previous study [30].

The propagation of wild-type SARS-CoV-2 (hCoV-19/Norway/Trondheim-S15/2020), human influenza A/Udorn/307/1972 (H3N2), HCV, and HIV-1, and engineering of recombinant mCherry-expressing SARS-CoV-2 strains (SARS-CoV-2-mCherry) have been described previously [25,30,31,32,33,34]. To quantitate the production of infectious virions, we titered the viruses using plaque assays or ELISA [25,30,31,32,33]. A plasmid harboring a subgenomic HEV sequence, coupled with a GFP reporter gene (Kernow-C1 p6 clone, gt3; GenBank Accession No. JQ679013) were used to generate HEV transcripts. Viral-capped RNAs were transcribed in vitro from a linearized plasmid using an mMESSAGE mMACHINE™ T7 Transcription Kit (Thermo Fisher, Waltham, MA, USA). A quantity of 1.5 × 10^7^ Huh-7.5 cells/mL in 400 µL of MaxCyte electroporation buffer were electroporated with 10 µg of p6-GFP subgenomic HEV RNA. Electroporation was carried out with a Gene Pulser system (Bio-Rad, Munich, Germany) and allowed to recover for 30 min in a 37 °C incubator. Recovered cells were resuspended in 10 mL prewarmed DMEM complete medium and maintained in an incubator for 24 h.

The lung organoids (LOs) were generated as described previously [35]. Briefly, induced pluripotent stem cells (IPSCs) were subjected to embryoid body induction using embryoid bodies (EB)/primitive streak media (10 μM Y-27632 and 3 ng/mL BMP4 in serum-free differentiation (SFD) media consisting of 375 mL Iscove’s Modified Dulbecco’s Medium (IMDM), 100 mL Ham’s F-12, 2.5 mL N2, 5 mL B27, 3.75 mL 7.5% BSA, 5 mL 1% penicillin–streptomycin, 5 mL GlutaMAX, 50 μg/mL ascorbic acid, and 0.4 μM monothioglycerol) in ultralow attachment plates. After 24 h, the media were replaced with endoderm induction media (10 μM Y-27632, 0.5 ng/mL BMP4, 2.5 ng/mL FGF2, and 100 ng/mL Activin A in SFD media). Extra media were added every day for 3 days. The embryoid bodies were collected and dissociated using 0.05% Trypsin/EDTA, and plated on fibronectin-coated plates with a cell density of 85,000 cells/cm^2^. Cells were then incubated in anteriorization media-1 (100 ng/mL Noggin, and 10 μM SB431542 in SFD media), followed by an incubation with anteriorization media-2 (10 μM SB431542, and 1 μM IWP2 in SFD media). The anteriorization media-2 were replaced with ventralization media (3 μM CHIR99021, 10 ng/mL FGF10, 10 ng/mL FGF7, 10 ng/mL BMP4, and 50 nM all-trans retinoic acid in SFD media), and incubated for two days. The cell monolayer was then lifted by gentle pipetting, and the suspended cells were transferred to an ultralow attachment plate where they would form the lung organoids.

### 2.2. Drug Testing and Drug-Sensitivity Quantification

Approximately 4 × 10^4^ Vero-E6 or Calu-3 cells were seeded per well in 96-well plates. The cells were grown for 24 h in DMEM or DMEM-F12, respectively, supplemented with 10% FBS and Pen–Strep. The medium was then replaced with DMEM or DMEM-F12 containing 0.2% BSA, Pen–Strep, and the compounds in 3-fold dilutions at 7 different concentrations. No compounds were added to the control wells. The cells were infected with SARS-CoV-2 or SARS-CoV-2-mCherry strains at an moi of 0.01 or mock. After 48 h of infection, drug efficacy on SARS-CoV-2-mCherry infected cells was measured on PFA- or acetone-fixed cells with fluorescence, as described previously [36]. After 72 h of infection, a CellTiter-Glo (CTG, Promega, Madison, WI, USA) and CellTox Green (CTxG, Promega) assays were performed to measure cell viability and cytotoxicity, respectively, as described previously [36,37,38].

For testing compound toxicity and efficacy against FluAV, approximately 4 × 10^4^ A549 cells were seeded in each well of a 96-well plate. The cells were grown for 24 h in DMEM, supplemented with 10% FBS and Pen–Strep. The medium was then replaced with DMEM containing 0.2% BSA, Pen–Strep, 0.5 μg/mL TPSK-trypsin, and compounds in three-fold dilutions at seven different concentrations. No compounds were added to the control wells. The cells were infected with FluAV (moi = 0.5) or mock. At 48 hpi, the medium was removed, and a CTG assay was performed to measure cell viability.

For testing compound toxicity and efficacy against HIV-1, approximately 4 × 10^4^ TZM-bl cells were seeded in each well of a 96-well plate in DMEM, supplemented with 10% FBS and Pen–Strep. The cells were grown for 24 h in a growth medium. The medium was then replaced with DMEM containing 0.2% BSA, Pen–Strep, and the compounds in 3-fold dilutions at 7 different concentrations. No compounds were added to the control wells. The cells were infected with HIV-1 (corresponding to 300 ng/mL of HIV-1 p24) or mock. At 48 h post-infection (hpi), the medium was removed from the cells, the cells were lysed, and firefly luciferase activity was measured using the Luciferase Assay System (Promega, Madison, WI, USA). In a parallel experiment, a CTG assay was performed to measure cell viability.

We also examined cytotoxicity and antiviral activity of drug combinations, using GFP-expressing HCV in Huh-7.5 cells by following previously described procedures [39].

For testing compound toxicity and efficacy against HEV, electroporated Huh-7.5 cells were seeded in the 384-well plate (3 × 10^3^ cells/well) with immune-modulators at indicated concentrations for 72 h. HEV replication was analyzed by determining the number of GFP-positive cells, using fully automated confocal microscopy (Operetta CLS; PerkinElmer Devices, Waltham, MA, USA).

The half-maximal cytotoxic concentration (CC_50_) for each compound was calculated based on viability/death curves obtained on mock-infected cells, after non-linear regression analysis with a variable slope using GraphPad Prism software version 7.0a. The half-maximal effective concentrations (EC_50_) were calculated based on the analysis of the viability of infected cells by fitting drug dose–response curves using a four-parameter (*4PL*) logistic function *f(x)*:(1)$$f\left(x\right)=A_{min}+\frac{A_{max}-A_{min}}{1+{\left(\frac{x}{m}\right)}^{\lambda }},$$
where *f(x)* is a response value at dose *x*, *A_min_* and *A_max_* are the upper and lower asymptotes (minimal and maximal drug effects), *m* is the dose that produces the half-maximal effect (EC_50_ or CC_50_), and *λ* is the steepness (slope) of the curve. The relative effectiveness of the drug was defined as the selectivity index (*SI* = CC_50_/EC_50_).

To quantify each drug response in a single metric, a drug sensitivity score (*DSS*) was calculated as a normalized version of the standard area under dose–response curve (*AUC*), with the baseline noise subtracted, and the normalized maximal response at the highest concentration (often corresponding to off-target toxicity):(2)$$DSS=\frac{AUC-t\left(x_{max}-x_{min}\right)}{\left(100-t\right)\left(x_{max}-x_{min}\right)\log_{10}Amin},$$
where activity threshold *t* equals 10%, and *DSS* is in the 0–50 range [40,41].

### 2.3. Drug Combination Testing and Synergy Calculations

Calu-3, A549, TZM-bl, or Huh-7.5 cells were treated with different concentrations of two drugs and infected with SARS-CoV-2-mCherry (moi = 0.01), FluAV (moi = 0.5), HIV-1 (corresponding to 300 ng/mL of HIV-1 p24), HCV, or mock. In addition, HEV-expressing Huh-7.5 cells were treated with different concentrations of two drugs. After 48 h, cell viability and reporter protein expression (SARS-CoV-2-mCherry, HIV-1, HCV-GFP, and HEV-GFP) were measured.

To test whether the drug combinations acted synergistically, the observed responses were compared with expected combination responses. The expected responses were calculated based on the ZIP reference model, using SynergyFinder version 2 [42,43]. Final synergy scores were quantified as average excess response due to drug interactions (i.e., 10% of cell survival beyond the expected additivity between single drugs represents a synergy score of 10). Additionally, we calculated most synergistic area scores for each drug combination—the most synergistic 3-by-3 dose-window in dose–response matrices.

LOs were treated with 0.5 μM remdesivir, 5 ng/mL IFNα2a, or their combination, and infected with SARS-CoV-2-mCherry (moi = 0.1). No compounds were added to the control wells. At 72 hpi, the LOs were stained using CTxG, and cell nuclei were stained with DAPI. Cells were fixed with PFA and imaged using an EVOS fluorescence microscope. Representative images (*n* = 3) were selected.

### 2.4. Gene Expression Analysis

Calu-3 cells or LOs were treated with drugs or vehicles, and infected with SARS-CoV-2-mCherry or mock. Total RNA was isolated using an RNeasy Plus Mini Kit (Qiagen, Hilden, Germany) from Calu-3 cells, or LOs. Polyadenylated mRNA was isolated from 250 ng of total RNA with a NEBNext Poly(A) mRNA Magnetic Isolation Module. A NEBNext Ultra II Directional RNA Library Prep Kit from Illumina was used to prepare samples for sequencing. Sequencing was done on a NextSeq 500 instrument (set up: single-end 1 × 76 bp + dual index 8 bp) using a NextSeq High Output 75-cycle sequencing kit (up to 400 M reads per flow cell). Reads were aligned using the Bowtie 2 software package version 2.4.2 to the NCBI reference sequence for SARS-CoV-2 (NC_045512.2) and to the Mesocricetus auratus MesAur1.0 assembly genome (https://ftp.ensembl.org/pub/release-100/fasta/mesocricetus_auratus/dna/, accessed on 1 May 2021). The number of mapped and unmapped reads that aligned to each gene were obtained with the featureCounts function from the Rsubread R-package version 2.10. The GTF table for the SARS-CoV-2 reference sequence was downloaded from https://ftp.ncbi.nlm.nih.gov/genomes/all/GCF/009/858/895/_009858895.2_SM985889v3/GCF_009858895.2_ASM985889v3_genomic.gtf.gz (accessed on 1 May 2021). The heatmaps were generated using the pheatmap package (https://cran.r-project.org/web/packages/pheatmap/index.html, accessed on 1 May 2021), based on log2-transformed or non-transformed profiling data. The clustered heatmaps were produced with the pheatmap v 1.0.12 R package, using either euclidean or minkowski distance functions (Raivo Kolde (2019). pheatmap: Pretty Heatmaps. R package version 1.0.12. https://cran.r-project.org/package=pheatmap, accessed on 1 May 2021).

### 2.5. Metabolic Analysis

Calu-3 cells or LOs were treated with drugs or vehicle, and infected with SARS-CoV-2-mCherry or mock. Metabolites were extracted from Calu-3 cells and LO supernatants, and analyzed as described [44]. The heatmaps were generated using the pheatmap package (https://cran.r-project.org/web/packages/pheatmap/index.html, accessed on 1 May 2021), based on log2-transformed or non-transformed profiling data. The clustered heatmaps were produced with the pheatmap v 1.0.12 R package, using either euclidean or minkowski distance functions (Raivo Kolde (2019). pheatmap: Pretty Heatmaps. R package version 1.0.12. https://cran.r-project.org/package=pheatmap, accessed on 1 May 2021).

## 3. Results

### 3.1. Type I IFNs Reduce SARS-CoV-2 Replication More Efficiently Than Type II and III IFNs in Cell Cultures

Although dexamethasone has been shown to improve survival of patients with severe or critical COVID-19 [45], there are currently no curative therapies against SARS-CoV-2. However, previous studies have uncovered several potent antiviral agents, including IFNs, against SARS-CoV-2 in vitro and in vivo [17,18,30]. Here, we tested type I, II, and III IFNs against wild-type SARS-CoV-2 (multiplicity of infection (moi) = 0.01) in Calu-3 and Vero-E6 cells, using cell viability and virus plaque reduction assays as readouts. We observed that at non-toxic concentrations, type I IFNs rescued both cell types from virus-mediated death, and reduced SARS-CoV-2 replication more efficiently than type II and III IFNs. However, the rescue was only partial, and virus replication was reduced only by 2–3 common logarithms (Figure 1).

To identify the type I IFN with the most activity against SARS-CoV-2 infection, we infected IFN-treated and untreated Calu-3 cells with SARS-CoV-2-mCherry (moi = 0.01) and collected media from the cells (p1) after 48 h. The media were diluted 25-fold and applied to noninfected cells for another 48 h (p2). Mock-infected cells were used as controls (Figure 2a). Fluorescence microscopy, fluorescence intensity analysis, and cell viability assay of p1 and p2 cells showed that IFNα1b, IFNα2a, and IFNw1 were more effective inhibitors of SARS-CoV-2 infection than IFNb1a. However, none of the IFNs tested were able to inhibit virus infection completely (Figure 2b–d).

Type I IFNs are encoded by multiple genes and vary slightly from one another in their protein structure. In basic research, IFNα2a is often used to elucidate the biological activities, structure, and mechanism of action of such type I IFNs. Thus, we next tested IFNα2a against various doses of SARS-CoV-2-mCherry and different times of drug addition. Calu-3 cells were treated with 1 μg/mL IFNα2a at indicated time points, then infected with SARS-CoV-2-mCherry at the indicated moi. After 48 h, fluorescence intensity and cell viability analyses were performed. We found that the efficacy of IFNα2a treatment in preventing SARS-CoV-2 infection was dependent on the viral load, decreasing in efficacy as the moi increased (Figure 3a). We also observed that antiviral efficacy was dependent on the time of addition, with increased efficacy when given prior to infection, rather than following infection (Figure 3b), which is in agreement with previous studies [46,47].

### 3.2. IFNα2a Reduces the SARS-CoV-2 RNA Synthesis and Promotes Virus-Mediated Induction of Type III IFNs, IFNb1, and ISGs in Calu-3 Cells

To shed new light on the mechanism of action of IFNα2a, we evaluated their effect on expression of cellular and viral RNA in mock- and SARS-CoV-2-infected Calu-3 cells. For this, cells were treated with 1 µg/mL of IFNα2a, other type I IFNs, or vehicle; then, they were infected with the virus or mock. After 24 h, we analyzed polyadenylated RNA using RNA-sequencing. We found that IFNα2a and other type I IFNs attenuated production of viral RNA (Figure 4a), while increasing expression of many ISGs in cells, regardless of virus- or mock-infection (Figure 4b). These ISGs include IFIT1, IFIT2, and IFIT3, which play a role in recognition of viral RNA; OASL and OAS2, which are involved in RNase L-mediated RNA degradation; and IDO1, which is essential for kynurenine biosynthesis [44,48,49,50]. Interestingly, IFNα2a and other type I IFNs boosted virus-activated expression of type III IFNs (IFNl1, IFNl2, IFNl3, and IFNl4), as well as IFNb1, which is a type I IFN. These results indicate that IFNα2a does not only trigger expression of ISGs, but also amplifies expression of other IFNs usually activated by viral infections, creating a positive feedback loop of IFN signaling during SARS-CoV-2 infection.

Next, we studied the effect of IFNα2a on the metabolism of mock- and SARS-CoV-2-infected Calu-3 cells. A total of 93 mainly polar metabolites were quantified at 24 hpi (Appendix A). We found that tyrosine and 4-hydroxyproline levels were substantially lowered during viral infection (log_2_FC < −2). Additionally, administration of IFNα2a or other type I IFNs lowered the levels of several metabolites, including tryptophan, while increasing kynurenine, regardless of viral infection (log_2_FC > 3; Figure 4c). This indicates that IFNα2a activates IDO1-mediated kynurenine biosynthesis, which could be associated with adverse reactions, such as suppression of T-cell responses, pain hypersensitivity, and behavior disturbance [51].

### 3.3. Synergistic IFNα2a-Based Combinations against SARS-CoV-2 Infection in Calu-3 Cells and Lung Organoids

Next, we examined whether IFNα2a in combination with known SARS-CoV-2 inhibitors remdesivir, EIDD-2801, camostat, cycloheximide, or convalescent serum [35,36,37,38,39,40,41,42,43,44,45,46,47,48,49,50,51,52] can protect cells from virus infection more efficiently and at lower concentrations than IFNα2a alone. Remdesivir and EIDD-2801 are nucleoside analogues, which inhibit viral RNA synthesis [53,54]. Camostat, a serine protease inhibitor, reduces SARS-CoV-2-cell membrane fusion by binding to host TMPRSS2 [55]. In addition, camostat possesses some potential beneficial immunomodulatory effects by interfering with the bradykinin/kallikrein pathways [56]. Cycloheximide inhibits translation elongation and, thereby, reduces SARS-CoV-2 replication [57]. Convalescent serum contains neutralizing antibodies that bind to the S protein of SARS-CoV-2, preventing virus entry into the cells [17].

We first confirmed the antiviral activities of these known viral inhibitors on Calu-3 cells using SARS-CoV-2-mCherry (Figure 5a and Appendix A). Then, we tested the antiviral efficacy and toxicity of these agents in combination with IFNα2a in Calu-3 cells by monitoring virus-mediated mCherry expression and cell viability (CTG). Each drug combination was tested in a 6 × 6 dose–response matrix, where 5 doses of single drugs were combined in a pairwise manner. As a result, we obtained dose–response matrices demonstrating virus inhibition and cell viability achieved by each combination (Figure 5c,d and Appendix A). We plotted synergy distribution maps, showing synergy (higher than expected effect) at each pairwise dose. For each drug pair, we calculated average ZIP synergy scores for the whole 6 × 6 dose–response matrix and for most synergistic 3 × 3 dose-regions, summarizing combination synergies into single metrics (Figure 5e). We observed that all combinations showed a strong synergy (synergy scores > 10) at various combination doses. Thus, the observed synergy allowed us to substantially decrease the concentration of both components to achieve antiviral efficacy that was comparable to those of individual drugs at high concentrations.

Both remdesivir and *rh*IFNα2a (Pegasys) have been approved for the treatment of COVID-19 infection in several countries. Because of their well-established roles in COVID-19 treatment, we decided to further validate the antiviral effect of this combination on iPSC-derived lung organoids (LOs). Thirty-day-old LOs were treated with 5 ng/mL IFNα2a, 0.5 μM remdesivir, or a combination thereof, then infected with SARS-CoV-2-mCherry. At 72 hpi, the organoids were analyzed for viral reporter-protein expression (mCherry), and cell death (CellTox Green). We found that IFNα2a–remdesivir substantially attenuated virus-mediated mCherry expression without affecting cell viability (Figure 6a). Virus titers were determined using plaque reduction assays (Figure 6b). The synergy score for a single dose of the IFNα2a–remdesivir combination was 9 (additive/synergistic effect), indicating that IFNα–remdesivir was able to attenuate SARS-CoV-2 protein expression and infectious virion production more efficiently than individual agents.

We also evaluated the effect of this combination treatment on viral and cellular RNA expression in LOs. RNA-sequencing revealed that at 48 hpi, IFNα2a–remdesivir reduced production of viral RNA to a substantially greater extent, compared to the effect of single agents (Figure 6c). Treatment with IFNα2a–remdesivir also led to elevated levels of ACE2 and other genes involved in lipid metabolism (APOA4, ADH4, CYP3A7, PON3, FADS6, SDR16C5, ENPP7, FABP2, CUBN, and SERPINA6) [58,59], for which transcription was substantially downregulated during SARS-CoV-2 infection (Appendix A and Figure 6d). Importantly, the set of IFNα2a-induced ISGs in LOs is consistent to what we observed in Calu-3 cells (Appendix A).

Furthermore, we studied the effect of IFNα2a–remdesivir on the metabolism of SARS-CoV-2- and mock-infected LOs. A total of 82 metabolites were quantified in LO culture supernatants at 48 hpi. Administration of IFNα2a–remdesivir prevented virus-mediated alteration of metabolism, excluding kynurenine biosynthesis (Appendix A and Figure 6e), which is in line with the results obtained on IFNα2a-stimulated Calu-3 cells, suggesting that this cocktail has high translatability.

### 3.4. Synergistic IFNα2a-Based Combinations against Other Viral Infections In Vitro

To extend our findings beyond SARS-CoV-2, we tested IFNα2a in combination with known HCV inhibitors, sofosbuvir and telaprevir, using GFP-expressing HCV in infected Huh-7.5 cells. Sofosbuvir is a nucleoside analogue, which inhibits viral RNA synthesis, whereas telaprevir is an orally available peptidomimetic that targets the HCV serine protease, and disrupts processing of viral proteins and formation of a viral replication complex. Eight different concentrations of the compounds alone or in combination were added to virus- or mock-infected cells. HCV-mediated GFP expression and cell viability were measured after 72 hpi to determine compound efficacy and toxicity. Both IFNα2a–sofosbuvir and IFNα2a–telaprevir lowered GFP expression without detectable cytotoxicity at indicated concentrations, with synergy scores of 3 and 5 (the most synergistic area scores: 14 and 16), respectively (Appendix A, Table 1).

Next, we studied IFNα2a in combination with known HEV inhibitors, NITD008 and ribavirin, against HEV infection in Huh-7.5 cells (Appendix A and Table 1). Both NITD008 and ribavirin are nucleoside analogs, which inhibit viral RNA synthesis. We observed that IFNα2a–NITD008 and IFNα2a–ribavirin were synergistic against HEV infection (ZIP synergy scores: 11 and 8; the most synergistic area scores: 14 and 19, respectively), while remaining nontoxic at synergistic doses for either drug.

We also tested IFNα2a in combination with known influenza inhibitor pimodivir against FluAV infection in A549 cells. Pimodivir (VX-787, JNJ-63623872) is an orally available anti-FluAV agent that targets viral polymerase basic protein 2, inhibits cap-snatching, and has shown promising results in Phase II clinical trials [60,61]. Cell viability was measured after 48 h in FluAV- and mock-infected cells to determine efficiency and toxicity of each compound and their combinations with IFNα2a (Appendix A and Table 1). We observed that IFNα2a–pimodivir was synergistic against FluAV infection (ZIP synergy score: 22, the most synergistic area score: 43), while remaining nontoxic at synergistic doses for either drug.

Finally, we tested IFNα2a in combination with known antiretroviral agent lamivudine against HIV-1 in TZM-bl cells. Lamivudine (3TC) is an orally available anti-HIV drug that inhibits viral reverse transcriptase [62]. Cell viability and HIV-induced luciferase expression were measured for each compound or their combination with IFNα2a after 48 h. We identified that treatment with IFNα2a and lamivudine was effective, while being nontoxic at synergistic drug concentrations, with ZIP synergy scores of 6 and a ZIP synergy score at the most synergistic area of 11 (Appendix A and Table 1).

## 4. Discussion

Currently, combinational therapies are still largely eschewed for the treatment of emerging viral infections in favor of monotherapies. This is in part due to the fact that many drug–drug interactions have not been fully explored or understood [63]. Here, we have reported several synergistic IFNα-based combination therapies that have better efficacy and lower toxicity than single drugs in vitro. We reported activities of IFNα2a combinations with remdesivir, EIDD-2801, camostat, cycloheximide, and convalescent serum against SARS-CoV-2. We also identified synergistic activity of IFNα2a with sofosbuvir or telaprevir against HCV infection, NITD008 or ribavirin against HEV infection, pimodivir against FluAV, as well as lamivudine against HIV-1 infection.

Based on our experiments, we propose the following mechanism of synergistic action for IFNα-based combinations (Figure 7). IFNα induces transcription of IFIT1, IFIT2, and IFIT3, as well as of OASL and OAS2, which recognize viral RNA and catalyze RNase L-mediated RNA degradation, respectively. Remdesivir, EIDD-2801, NITD008, ribavirin, sofosbuvir, pimodivir, and lamivudine, also inhibit viral RNA transcription [53,54,60,64]. Therefore, these antiviral agents and IFNα target the same step of viral replication to achieve synergy. Similarly, other IFNα-induced ISGs, together with inhibitors, can synergistically target the same processes, which play a critical role in viral replication. This is in agreement with previous studies [65,66,67].

Interestingly, it has been noted that some patients with severe symptoms can develop autoantibodies targeting their own IFNs, presenting a subset of patients for whom rhIFNs may be beneficial [68,69]. Therefore, the administration of IFN-based combinations may not only act as a potent antiviral treatment regimen, but may also actively counteract an immune-evading mechanism used by viruses and boost host immunity.

It is important to note that for further in vivo studies, it will be necessary to consider the time and route of administration, as well as the metabolism of IFNα-containing drug combinations. In particular, intranasal IFNα treatment is beneficial only when administered before the onset of clinical signs in the SARS-CoV-2 hamster model [70]. In addition, earlier intramuscular administration of IFN-α was suggested to be beneficial for COVID-19 patients (10.2217/fvl-2020-0404). Moreover, differences in biological activity and structure between different type I IFNs may reduce the transcription of ISGs, and eliminate the synergistic effect of the combinational treatment [71]. Therefore, the overall antiviral effect for treatment with IFNs in vivo could be dampened. Further studies will be required to determine if the full synergism with IFNα-based combinations that was witnessed for in vitro models could be translated in vivo. Despite these difficulties, we believe the identification of this combinational therapy illustrates an important approach to the development of effective antiviral treatment options.

## 5. Conclusions

Our proposed IFNα-based drug combinations leverage an approach that balances the bolstering of innate antiviral immunity of the host and supplemental external antiviral activity through the administration of synergistic antiviral agents. Furthermore, combinational antiviral treatments, which reduce viral replication at lower concentrations than are required with monotherapies, are valuable due to their lowered potential for adverse effects and improved outcomes.

## Figures and Tables

**Figure 1 viruses-13-02489-f001:**
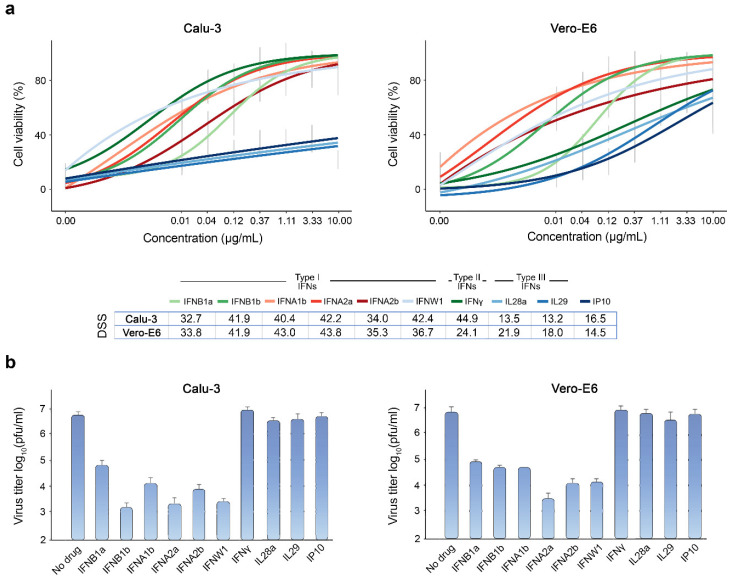
Type I IFNs rescue Calu-3 and Vero-E6 cells from SARS-CoV-2-mediated death and attenuate virus replication. (**a**) The effect of different doses of IFNs on viability of SARS-CoV-2-infected (moi = 0.01) Calu-3 and Vero-E6 cells. Cell viability was determined using the CTG assay at 72 hpi. Mean ± SD; *n* = 3. The anti-SARS-CoV-2 activity of the IFNs was quantified using drug sensitivity scores (DSS). (**b**) The effects of IFNs on viral replication, measured by plaque reduction assay. Mean ± SD; *n* = 3. Statistical testing was performed using pairwise Wilcoxon Rank Sum Tests and the results can be found in Appendix A.

**Figure 2 viruses-13-02489-f002:**
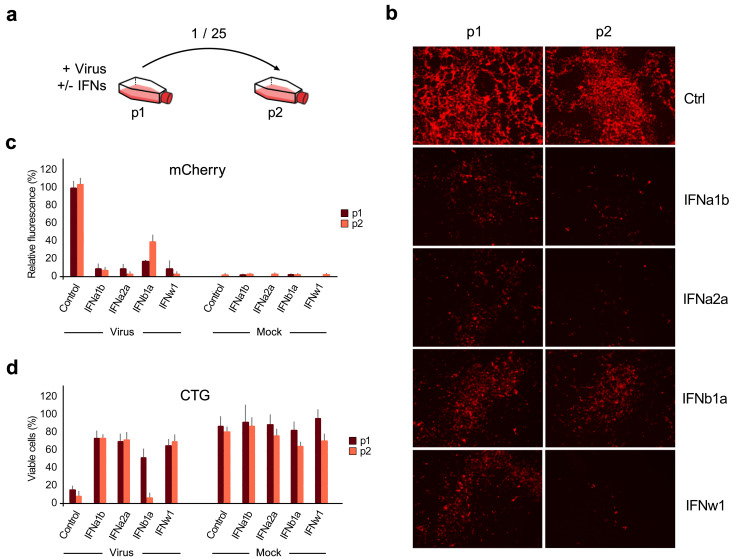
IFNα1b, IFNα2a, and IFNw1 are more effective than IFNb1a against SARS-CoV-2-mCherry infection in Calu-3 cells. (**a**) Schematic representation of the experimental setup. (**b**) Fluorescent images of non-treated (Ctrl) and IFN-treated (1 μg/mL) SARS-CoV-2-mCherry-infected (moi = 0.01) Calu-3 cells (p1) and cells (p2) treated with 25-fold diluted media from P1 cells taken at 48 hpi. (**c**,**d**) Fluorescence intensity and cell viability analyses of p1 and p2 cells at 48 hpi. Mock-infected cells were used as controls (Mean ± SD; *n* = 3). Statistical testing was performed using pairwise Wilcoxon Rank Sum Tests and the results can be found in Appendix A.

**Figure 3 viruses-13-02489-f003:**
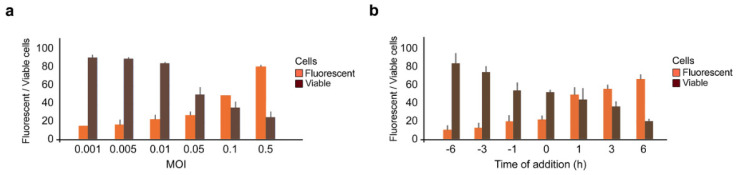
Antiviral activity of IFNα2a depends on moi of SARS-CoV-2-mCherry and time of drug administration. (**a**) Calu-3 cells were treated with 1 μg/mL IFNα2a and infected with indicated moi of SARS-CoV-2-mCherry. Fluorescence intensity and cell viability were measured after 48 h (Mean ± SD; *n* = 3). (**b**) Calu-3 cells were treated with 1 μg/mL IFNα2a prior, simultaneously, or post infection with SARS-CoV-2-mCherry (moi = 0.01). Fluorescence intensity and cell viability were measured after 48 h (Mean ± SD; *n* = 3).

**Figure 4 viruses-13-02489-f004:**
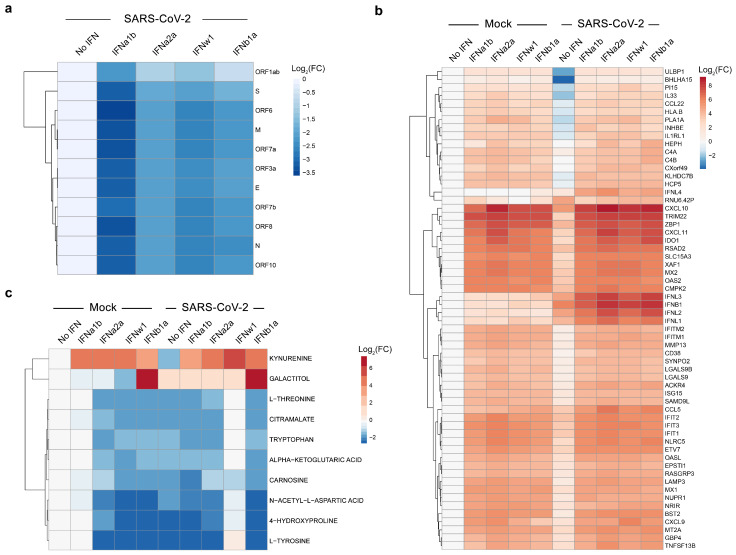
Transcriptomic and metabolomic analyses of mock- and SARS-CoV-2-infected Calu-3 cells, non-treated or treated with type I IFNs. (**a**) Calu-3 cells were stimulated with IFNs (1 μg/mL) or non-stimulated, and infected with SARS-CoV-2 (moi = 0.01). A heatmap of viral RNAs affected by treatment is shown. Each cell is colored according to the log2-transformed expression values of the samples, expressed as fold-change relative to the non-treated control. (**b**) Calu-3 cells were either stimulated with recombinant human IFN (1 μg/mL) or left untreated with IFN, then infected with either mock or SARS-CoV-2 (moi = 0.01). A heatmap of the most variable cellular genes affected by treatment and virus infection is shown. Each cell is colored according to the log2-transformed expression values of the samples, expressed as fold-change relative to the non-treated, mock-infected control. (**c**) Cells were treated as for panel b. After 24 h, the cell culture supernatants were collected, and metabolite levels were determined by LC-MS/MS. A heatmap of the most-affected metabolites is shown. Each cell is colored according to the log2-transformed profiling values of samples, expressed as fold-change relative to the mock control.

**Figure 5 viruses-13-02489-f005:**
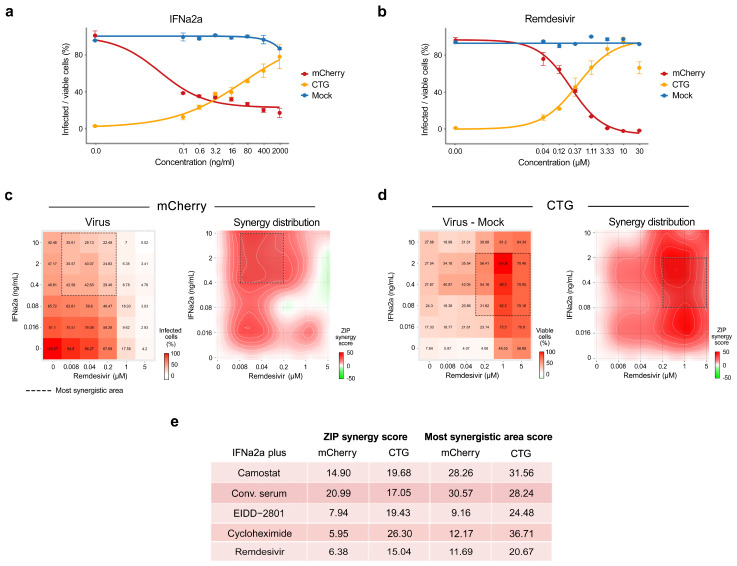
Synergistic IFNα2a-based combinations against SARS-CoV-2-mCherry infection in Calu-3 cells. (**a**,**b**) Calu-3 cells were treated with increasing concentrations of IFNα2a or remdesivir, and infected with the SARS-CoV-2-mCherry or mock. After 48 h, the virus-mediated mCherry expression was measured (red curves). After 72 h, viability of virus- and mock-infected cells was determined using a CTG assay (yellow and blue curves, respectively). Mean ± SD; *n* = 3. (**c**) The 6 × 6 dose–response matrices and interaction landscapes of IFNα2a and remdesivir obtained using fluorescence analysis of SARS-CoV-2-mCherry-infected Calu-3 cells. ZIP synergy score was calculated for the drug combinations. (**d**) The 6 × 6 dose–response matrices and interaction landscapes of IFNα2a and remdesivir obtained using a cell viability assay (CTG) on mock- and SARS-CoV-2-mCherry-infected Calu-3 cells. The selectivity for the indicated drug concentrations was calculated (selectivity = efficacy-(100-Toxicity)). ZIP synergy scores were calculated for indicated drug combinations. (**e**) ZIP synergy scores (synergy score for whole 6 × 6 dose–response matrix) and the most synergistic area scores (synergy score for most synergistic 3 × 3 dose-regions) calculated for indicated drug combinations.

**Figure 6 viruses-13-02489-f006:**
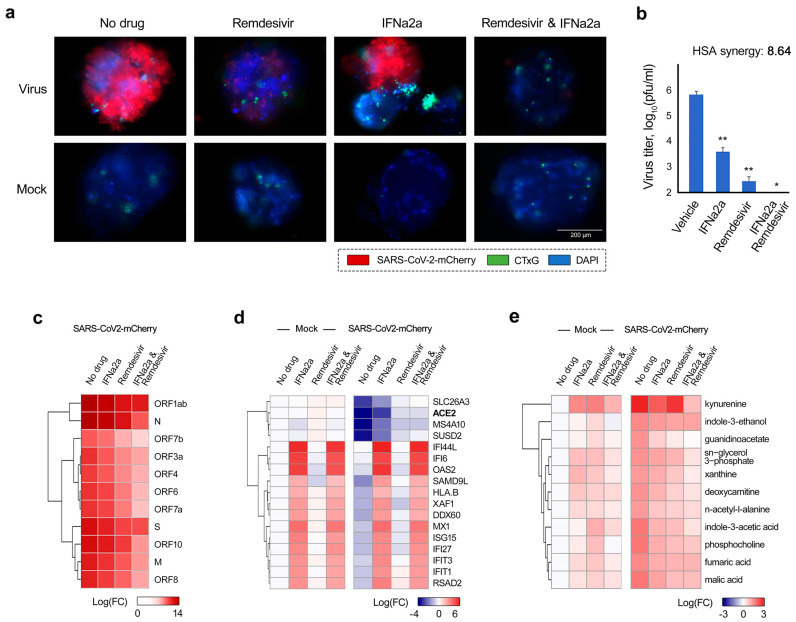
Antiviral effect of IFNα2a–remdesivir combination in human lung organoids (LOs). (**a**) LOs were treated with 0.5 μM remdesivir, 5 ng/mL IFNα2a, their combination, or vehicle, and infected with SARS-CoV-2-mCherry (moi = 0.1) or mock. Fluorescence of drug- or carrier-treated SARS-CoV-2-mCherry-infected LOs was detected at 72 hpi. Virus infection, cell nuclei, and cytotoxicity are shown in red, blue, and green, respectively. Scale bars, 200 μm. (**b**) The effects of IFNα–remdesivir combination on viral replication in LOs, measured by plaque reduction assay. Mean ± SD; *n* = 3. Statistical significance is indicated with asterisks (* *p* < 0.05, ** *p* < 0.1, Wilcoxon test), the significance levels were compared to the vehicle. Synergy score for single dose of IFNα–remdesivir combination was calculated as an excess combinatorial effect over expected effect, given by the HSA synergy model. (**c**) LOs were treated with 0.5 μM remdesivir, 5 ng/mL IFNα2a, their combination, or vehicle, and infected with SARS-CoV-2-mCherry (moi = 0.1). After 48 h, total RNA was extracted and sequenced. A heatmap of viral RNAs affected by treatment is shown. Each cell is colored according to the log2-transformed expression values of the samples, expressed as log2 fold-change relative to the non-treated control. (**d**) LOs were treated and infected as for panel a. After 48 h, total RNA was extracted and sequenced. A heatmap of the most variable cellular genes affected by treatment and virus infection is shown. Each cell is colored according to the log2-transformed expression values of the samples, expressed as fold-change relative to the non-treated, mock-infected control. Cut-off—3.75. (**e**) Cells were treated as for panel a. After 48 h, the cell culture supernatants were collected, and metabolite levels were determined by LC-MS/MS. A heatmap of the most-affected metabolites is shown. Each cell is colored according to the log2-transformed profiling values of samples, expressed as fold-change relative to the mock control. Cut-off—1.5.

**Figure 7 viruses-13-02489-f007:**
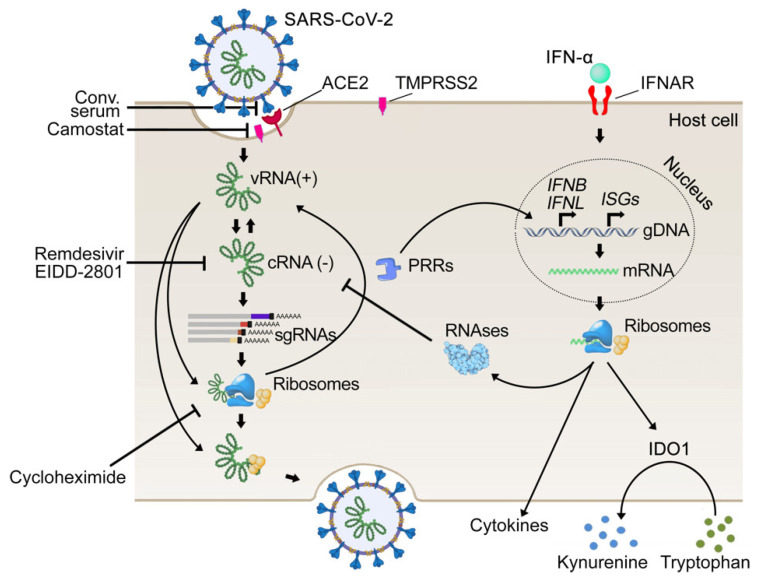
Schematic representation of mechanisms of anti-SARS-CoV-2 actions of remdesivir, EIDD-2801, camostat, cycloheximide, and convalescent serum, and stages of virus replication cycle they target.

**Table 1 viruses-13-02489-t001:** Synergy and the most synergistic area scores of IFNα2a-based combinations tested against HCV, HEV, FluAV, and HIV infections in cell cultures.

Virus	Cell Line	IFNα2a Plus	ZIP Synergy Score	Most Synergistic Area Score
HCV	Huh-7.5	Sofosbuvir	3.2	13.7
		Telaprevir	5.0	15.7
HEV	Huh-7.5	NITDD008	7.8	18.7
		Ribavirin	10.7	14.4
FluAV	A549	Pimodivir	22.1	42.8
HIV	TZM-bl	Lamivudine	5.7	11.4

## Data Availability

All data generated or analyzed during this study are included in this published article and its Appendix A.

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
