# Peer review of "Synergistic Interferon-Alpha-Based Combinations for Treatment of SARS-CoV-2 and Other Viral Infections"

_viruses, 2021, doi:10.3390/v13122489_

Round 1

Reviewer 1 Report

All the questions were answered. I have no other concerns about the quality and content of the paper.

Author Response

Thank you.

Reviewer 2 Report

The authors addressed most of the comments. However, there are still some minor issues. 

One of the microM symbol is typed wrong in Fig. 6 legend.

Legends of figure 7 seems to be deleted including its mention in the manuscript. Was it intentional? If this figure is now removed from the manuscript, there is still mention of lung of Syrian hamsters in line 271. I would suggest to show the data, even if they do not show synergy. Authors can conclude as discussed in the rebuttal letter indicating that there are variabilities in lung organoid and in vivo model. This always happens and understandable.

The discussion is short, the authors can discuss role of interferons as indicated by others including using Syrian hamster model (PMID: 34370799) or mouse models of coronavirus infection (PMID: 34705554) where animals were treated with interferons either prophylactically or therapeutically.

Author Response

One of the microM symbol is typed wrong in Fig. 6 legend.

Re: Corrected.

Legends of figure 7 seems to be deleted including its mention in the manuscript. Was it intentional? If this figure is now removed from the manuscript, there is still mention of lung of Syrian hamsters in line 271. I would suggest to show the data, even if they do not show synergy. Authors can conclude as discussed in the rebuttal letter indicating that there are variabilities in lung organoid and in vivo model. This always happens and understandable.

Re: We have removed the figure and the corresponding sections from the text because this was a pilot experiment with prophylactic single-dose treatment of SARS-CoV-2 infected hamsters. Further experiments are needed to conclude that IFNa and remdesivir have an additive or synergistic effect in this model system.

The discussion is short, the authors can discuss role of interferons as indicated by others including using Syrian hamster model (PMID: 34370799) or mouse models of coronavirus infection (PMID: 34705554) where animals were treated with interferons either prophylactically or therapeutically.

Re: We expanded the discussion accordingly: "It is important to note that for further in vivo studies, it is necessary to take into account the time and route of administration, as well as the metabolism of IFNa-containing drug combinations described in our in vitro study. In particular, intranasal IFNa treatment is beneficial only when administered before clinical signs onset in the SARS-CoV-2 hamster model {Bessiere, 2021 #39}. Also earlier intramuscular administration of IFN-α was suggested to be beneficial for COVID-19 patients (10.2217/fvl-2020-0404). 

Reviewer 3 Report

The manuscript described IFNa-based combination therapies due to the urgent need for new antivirals with intended efficacy and tolerable side effects. Combinational therapies can be an attractive target for further pre-clinical and clinical development against emerging and re-emerging viral infections.  Investigating the synergistic effect of the combinational therapy has been performed using cell and organoid model system by observing the transcriptome and metabolomic changes and evaluating the impact on viral RNA transcription, protein synthesis and processing, and in vivo hamster model only for quantification of virus load in lungs.

Why Pen-Strep combination used while testing compounds as no compounds were added to control? Why keeping PS in the medium is required after successful cell passage? Does this application not affect the global gene expression/metabolomic pattern?

Figure 6b, 7b, and 7d have a confusing schematic representation of significance levels? What are the significance levels compared to? All to the vehicle or other combinations?

How is the hierarchical clustering calculated in the heat maps of transcriptomic and metabolomic results?

Line 149 The methodology details of lung organoid preparation have been referenced. However, this is confusing while reading that the study focused only on in vitro models; however, a hamster model is also mentioned further in the text. Including this detail has been neglected and causing confusion. The section needs clarification by adding extra sentences confirming that similar hamsters were used in vivo or performed differently.

It is also confusing that each research section has been used a different set of cells. Hence, it looks unorganized and becomes complicated for the readers to follow up and get a comprehensive comparative overview of this detailed set of analyses. It will be helpful to have a graphical abstract clarifying the overall design, or a tablet will ease the follow-up.

Is there any reference for drug testing and sensitivity calculation, or is it new for this study or used package program has contains the reference?

Line 232 what is the “DSS ∈ [0,50].”

Line 331 What is the proof of concept behind “(p2) treated with 25-fold diluted media from P1 cells taken at 48 hpi.”

Expression analyses are discussed for cells and LOs. Are there any results from in vivo samples from hamsters?

The expression analysis demonstrates ISGs. What about ribosomal genes etc.? Is there any preselection applied in the analysis?

Author Response

1. Why Pen-Strep combination used while testing compounds as no compounds were added to control? Why keeping PS in the medium is required after successful cell passage? Does this application not affect the global gene expression/metabolomic pattern?

Re: We added Pen-Strep to both untreated and drug-treated cells. We used PS during the isolation of wild-type SARS-CoV-2 in VERO-E6 and in all other experiments, and therefore we believe that these antibiotics should not interfere with the results of our efficacy tests. 

2. Figure 6b, 7b, and 7d have a confusing schematic representation of significance levels? What are the significance levels compared to? All to the vehicle or other combinations?

Re: We removed the fig. 7 (in vivo test) during the previous revision, because it showed preliminary results that need to be confirmed. We apologize for the rice confusion in fig 6b legend and now clarified the explanations for the figure as follows: "Statistical significance is indicated by asterisks (* p <0.05, ** p <0.1, Wilcoxon test), significance levels were compared to vehicle."

3. How is the hierarchical clustering calculated in the heat maps of transcriptomic and metabolomic results?

Re: We have added to the methods section a sentence describing how we constructed clustered heatmaps: “Clustered heatmaps were generated with pheatmap v 1.0.12 R using Euclidean or Minkowski distance functions (Raivo Kolde (2019). Pheatmap: Pretty Heatmaps Package . R version 1.0.12. Https://cran.r-project.org/package=pheatmap) "

4. Line 149 The methodology details of lung organoid preparation have been referenced. However, this is confusing while reading that the study focused only on in vitro models; however, a hamster model is also mentioned further in the text. Including this detail has been neglected and causing confusion. The section needs clarification by adding extra sentences confirming that similar hamsters were used in vivo or performed differently.

Re: We have omitted Fig. 7 (the efficacy experiment in hamsters) during previous revision, because we only ran the experiment once..

4. It is also confusing that each research section has been used a different set of cells. Hence, it looks unorganized and becomes complicated for the readers to follow up and get a comprehensive comparative overview of this detailed set of analyses. It will be helpful to have a graphical abstract clarifying the overall design, or a tablet will ease the follow-up.

Re: Now we have transformed Fig. 7 in Table 1 and indicated the cell lines that we used for the analysis.

5. Is there any reference for drug testing and sensitivity calculation, or is it new for this study or used package program has contains the reference?

Re: We apologize for the confusion, and have now added a link to the Breeze publication (https://doi.org/10.1093/bioinformatics/btaa138) that implements drug sensitivity score (DSS) calculations and DSS original publication (https://doi.org/10.1038/srep05193).

6. Line 232 what is the “DSS ∈ [0,50].”

Re: We have now rephrased the sentence as following: "DSS belongs to [0,50] range".

7. Line 331 What is the proof of concept behind “(p2) treated with 25-fold diluted media from P1 cells taken at 48 hpi.”

Re: With this experiment, we have demonstrated that IFNs cannot completely suppress the replication of SARS-CoV-2-mCherry.

8. Expression analyses are discussed for cells and LOs. Are there any results from in vivo samples from hamsters?

Re: We deleted our preliminary results on hamsters in the previous revision.

9. The expression analysis demonstrates ISGs. What about ribosomal genes etc.? Is there any preselection applied in the analysis?

Re: Supplemental data contains all genes affected by treatment, viral infection, or both.

This manuscript is a resubmission of an earlier submission. The following is a list of the peer review reports and author responses from that submission.

Round 1

Reviewer 1 Report

In this papers, the Authors present results about the effectiveness of drugs in the treatment of Covid-19 infection. In vitro and in vivo data are presented. Synergistic effects are also discussed.

The paper is well written and results are nicely presented.

However, some points need to be clarified, to improve the quality of the paper.

1) The authors do not discuss whether thy check the cell uptake of the molecules/formulations under investigation. Since the administration to the animals is quite different, this should be clarified.

2) The viral protein fused to mCherry (or the nature of labeled viral components) should be explicitly mentioned.

3) The part about polyadenylated mRNA needs more details (what mRNA is detected, etc)

4) What kind of cytotoxicity assay is performed?

5) Fig 7, panel A is hard to read

6) Fig 6, panel A: the image has a poor quality

7) What kind of microscopy is performed? the images don't seem to come from a confocal microscope.

Author Response

  1. We have now discussed this and other issues associated with observed in vitro and in vivo differences: "The anti-SARS-CoV-2 activities of IFNa-remdesivir  vary between human LOs and hamsters. We were able to show that this treatment suppressed viral RNA expression more effectively than the drugs alone in LOs. The synergism demonstrated in the LOs was reduced to an additive effect in our hamster model. This observation is in consensus with the fact that while in vitro treatment with IFNa-remdesivir was successful at inducing transcription of antiviral genes, no such activity was observed in the hamster model of treatment of infection. Differences in biological activity and structure between the IFNs of Syrian hamsters and those of murine rodents [59] may have dampened the overall antiviral effect for IFNa treatment, thereby reducing the transcription of ISGs and eliminating the synergistic effect of the combinational treatment. Further studies will be required to determine if the full synergism that was witnessed for in vitro models could be translated into animal models as well. In addition the cell uptake of the molecules/formulations under investigation is quite different from the administration to the animals."
  2. The construction of SARS-CoV-2-mCherry virus was described previously (PMID: 33630831). mCherry was fused to ORF7a-2A. "The propagation of wild-type SARS-CoV-2 (hCoV-19/Norway/Trondheim-S15/2020) and engineering of recombinant mCherry-expressing SARS-CoV-2 strains (SARS-CoV-2-mCherry) have been described previously [24, 29-33]. "
  3. We have now added the details: "Polyadenylated mRNA was isolated from 250 ng of total RNA with NEBNext Poly(A) mRNA magnetic isolation module. NEBNext Ultra II Directional RNA Library Prep kit from Illumina was used to prepare samples for sequencing. Sequencing was done on NextSeq 500 instrument (set up: single-end 1 x 76 bp + dual index 8 bp) using NextSeq High Output 75 cycle sequencing kit (up to 400M reads per flow cell). Reads were aligned using the Bowtie 2 software package version 2.4.2 to the NCBI reference sequence for SARS-CoV-2 (NC_045512.2) and to the Mesocricetus auratus MesAur1.0 assembly genome (https://ftp.ensembl.org/pub/release-100/fasta/mesocricetus_auratus/dna/). The number of mapped and unmapped reads that aligned to each gene were obtained with the featureCounts function from Rsubread R-package version 2.10. The GTF table for the SARS-CoV-2 reference sequence was downloaded from https://ftp.ncbi.nlm.nih.gov/genomes/all/GCF/009/858/895/GCF_009858895.2_SM985889v3/GCF_009858895.2_ASM985889v3_genomic.gtf.gz. The heatmaps were generated using the pheatmap package (https://cran.r-project.org/web/packages/pheatmap/index.html) based on log2-transformed or non-transformed profiling data."
  4. CellTiter-Glo (CTG, Promega) and CellTox Green (CTxG, Promega) assays were performed to measure cell viability and cytotoxicity, respectively, as described [35-37].

  5. Fig. 7 is now enlarged and has higher resolution.

  6. Fig. 6 is now enlarged and has higher resolution.

  7. Cells were fixed with PFA and imaged using EVOS fluorescence microscope.

Many thanks for your comments.

Reviewer 2 Report

This study by Ianevski et al aimed to investigate the effectiveness of combination therapy combined with type I interferon therapy. They report a synergistic inhibition of SARS-CoV-2 by remdesivir along with novel antiviral agents when combined with type I interferons. They utilized both in vitro and in vivo models of infection to test the therapeutic synergy between combination therapies. This is quite timely and urgently needed. However, there are some concerns regarding the data presented and conclusions drawn:

Strengths:

Clinical importance of topic

Characterizing a wide range of interferons.

Major concerns:

The major concern is interpretation of in vivo data in hamsters. Fig. 7b and 7d shows do not show any statistical decrease in the viral load in the combination group compared to single treatment group. Which is key in showing additive or synergistic effects.

The pretreatment of hamsters shows that there are limitations of translating these findings for therapeutic approaches.

The interferon concentrations used at 1ug/ml seems to be too high for in vitro experiments and what is the justification for using such as high concentrations. Compared to that, authors used 5ng/ml when treating lung organoids.  

The inhibitory effects should be investigated in a therapeutic settings both in vivo and in vitro (post viral infection).

Other comments:

Statistics is missing in Fig. 1 and 2

The expression of ISGs in Fig. 4. including other interferons by interferon treatment is well known phenomenon and does not constitute novel findings.

Author Response

  1. To quantify synergistic effect between drugs, we used an HSA synergy model that calculates synergy as an excess over the maximum single drug response (https://doi.org/10.1093/nar/gkaa216). Other popular synergy models (e.g. ZIP, or Loewe) are not applicable for calculating synergy at a single combination dose pair. We indeed found that combination was not synergistic, but rather additive, HSA synergy score is equal to 1. We now better explained this in the manuscript.We now discuss the limitation of our study: " The anti-SARS-CoV-2 activities of IFNa-remdesivir  vary between human LOs and hamsters. We were able to show that this treatment suppressed viral RNA expression more effectively than the drugs alone in LOs. The synergism demonstrated in the LOs was reduced to an additive effect in our hamster model. This observation is in consensus with the fact that while in vitro treatment with IFNa-remdesivir was successful at inducing transcription of antiviral genes, no such activity was observed in the hamster model of treatment of infection. Differences in biological activity and structure between the IFNs of Syrian hamsters and those of murine rodents [61] may have dampened the overall antiviral effect for IFNa treatment, thereby reducing the transcription of ISGs and eliminating the synergistic effect of the combinational treatment. Further studies will be required to determine if the full synergism that was witnessed for in vitro models could be translated into animal models as well. In addition the cell uptake of the molecules/formulations under investigation is quite different from the administration to the animals. " 
  2. We agree, that the pretreatment of hamsters is a limitation of translating these findings for therapeutic approaches. However, "Despite these limitation, we believe the identification of this combinational therapy illustrates an important approach to the development of effective treatment options against SARS-CoV-2 infection. Overall, combinational antiviral treatments, which reduce viral replication at lower concentrations than are required with monotherapies, are valuable due to their lowered potential for adverse effects and potential improved outcomes. Further development of these treatments, especially our proposed IFNa-remdesivir combination and others which incorporate IFNs to mobilize the body’s own intrinsic antiviral immunity, should be pursued".
  3. We agree that  interferon concentrations used for cell culture experiments  were high (1ug/ml), but not cytotoxic. Same concentration could be used in LOs, however 5ng/ml was also effective. 
  4. Indeed, inhibitory effects could be investigated in a therapeutic settings both in vivo and in vitro (post viral infection). However, this is a pilot study, and in our other studies we devote attention for more effective synergistic combinations (such as Pegasys-nafamostat).
  5. We run the pairwise Wilcoxon Rank Sum Test and added a supplementary file with the results (Table S2 and S3). We also mentioned it in the manuscript.
  6. Indeed, the expression of ISGs in Fig. 4. including other interferons by interferon treatment is well known phenomenon and does not constitute novel findings. However, the differential effects of these IFNs on SARS-CoV-2 infected cells is novel.

Many thanks for your comments!